

# Increased tooth brushing frequency is associated with reduced gingival pocket bacterial diversity in patients with intracranial aneurysms

Mikko J. Pyysalo[1,2,3,4], Pashupati P. Mishra[5,6], Kati Sundström[5,6], Terho Lehtimäki[5,6], Pekka J. Karhunen[5,6] and Tanja Pessi[4]

[1] Department of Otorhinolaryngology, Faculty of Medicine and Health Technology, University of Tampere, Tampere, Finland
[2] Department of Oral and Maxillofacial diseases, Tampere University Hospital, Tampere, Finland
[3] Oral Health Services, City of Tampere, Tampere, Finland
[4] Department of Molecule Microbiology, Faculty of Medicine and Health Technology, University of Tampere, Tampere, Finland
[5] Faculty of Medicine and Health Technology and Finnish Cardiovascular Research Center, University of Tampere, Tampere, Finland
[6] Fimlab Laboratories ltd, Tampere, Finland

Corresponding author
Mikko J. Pyysalo,
mikko.pyysalo@fimnet.fi

## ABSTRACT

**Objectives.** The objective of this study was to investigate the association of tooth brushing frequency and bacterial communities of gingival crevicular fluid in patients subjected to preoperative dental examination prior to operative treatment for unruptured intracranial aneurysms.

**Methods.** Gingival crevicular fluid samples were taken from their deepest gingival pocket from a series of hospitalized neurosurgical patients undergoing preoperative dental screening ($n = 60$). The patients were asked whether they brushed their teeth two times a day, once a day, or less than every day. Total bacterial DNA was isolated and the V3–V4 region of the 16S rRNA gene was amplified. Sequencing was performed with Illumina's 16S metagenomic sequencing library preparation protocol and data were analyzed with QIIME (1.9.1) and R statistical software (3.3.2).

**Results.** Bacterial diversity (Chao1 index) in the crevicular fluid reduced along with reported tooth brushing frequency ($p = 0.0002$; $R2 = 34\%$; $p$ (adjusted with age and sex) $= 0.09$; $R2 = 11\%$) showing that patients who reported brushing their teeth twice a day had the lowest bacterial diversity. According to the differential abundant analysis between the tooth brushing groups, tooth brushing associated with two phyla of fusobacteria [$p = 0.0001$; $p = 0.0007$], and one bacteroidetes ($p = 0.004$) by reducing their amounts.

**Conclusions.** Tooth brushing may reduce the gingival bacterial diversity and the abundance of periodontal bacteria maintaining oral health and preventing periodontitis, and thus it is highly recommended for neurosurgical patients.

## INTRODUCTION

The human oral cavity is considered healthy when the oral microflora is composed of indigenous bacteria and is properly balanced (*Socransky & Haffajee, 2002*).

When dental plaque biofilm develops, it causes gingival inflammation that leads to marginal gingival swelling, the initiation of pocket formation, and the increased exudation of gingival crevicular fluid. If undisturbed, a mature biofilm develops in 1 to 2 weeks (*Marsh & Martin, 2009*). Dental plaque biofilms are composed of microcolonies of bacterial cells that are non-randomly distributed in a shaped matrix or glycocalyx (*Socransky & Haffajee, 2002*). Six bacterial groups are considered to be early biofilm colonizers: *Actinomyces*, a yellow complex comprising members of the genus *Streptococcus*, a green complex comprising the *Capnocytophaga* species, *Actinobacillus actinomycetemcomitans* serotype a, *Eikenella corrodens* and *Campylobacter concisus,* and a purple complex comprising *Veillonella parvula* and *Actinomyces odontolyticus* (*Socransky et al., 1998*). The growth of these six groups usually precedes the colonization of gram-negative orange (*Fusobacterium nucleatum, Prevotella intermedia, Prevotella nigrescens, Parvimonas micra, Streptococcus constellatus, Eubacterium nodatum, Campylobacter showae, Campylobacter gracilis, Campylobacter rectus*) and red (*Porphyromonas gingivalis, Tannerella forsythia, Treponema denticola*) complexes (*Socransky et al., 1998*). Recent studies have shown that the microbiota associated with periodontitis is much more complex than previously understood. Species from the genera *Parvimonas, Filifactor, Dialister, Granilucatella*, and *Synergistes* are found together with species belonging to the orange and red complexes (*Colombo et al., 2009*). Despite the active biofilm research, it still remains unclear which subgingival bacterial profiles are associated with the development of periodontal disease (*Kononen & Muller, 2014*).

The use of cultivation independent 16S rRNA gene sequencing provides a powerful tool to analyze the complex microbiomes of environmental and clinical samples. Ai and co-workers (*Ai et al., 2017*) have suggested that alterations in the subgingival microbiome can be identified using metagenomic sequencing and used as a predictive marker of early periodontitis. The aim of this study was to investigate the impact of tooth brushing on the bacterial communities in the gingival crevicular fluid of patients undergoing preoperative dental screening prior to intracranial aneurysm treatment.

## MATERIALS AND METHODS

In this study, the STROBE checklist for cohort studies was followed where applicable. This study was approved by the Ethics Committee of Pirkanmaa Hospital District (code R12217). Sixty patients subjected to preoperative dental screening due to a planned neurosurgical procedure for unruptured intracranial aneurysm were recruited to the study between September 2012 and December 2014. The inclusion criteria were having a saccular aneurysm, being aged between 18 and 99 years, having the ability to give informed consent, and having a clinical condition that allowed safe transportation to the radiological department situated in the same building. An informed consent form approved by the

hospital ethical committee was obtained from all of the study patients. The exclusion criteria were requiring intensive care and being unable to give informed consent.

During the acute hospitalization period for aneurysm treatment, an experienced oral and maxillofacial surgeon (MP) clinically investigated the patients' teeth. Periodontal probing was carried out using a standard WHO periodontal probe (LM 8-550B Si, LM-instruments Ltd, Parainen, Finland) with about 20 g force. Gingival pockets were measured from 6 sites of each tooth, and crevicular fluid samples for bacterial DNA analyses were taken from the deepest gingival pocket using a sterile blotting paper pin (Pearl Dent Co, Ho Chi Minh City, Vietnam). The samples were refrigerated at −80 °C within 15 min. Patients were asked to report their tooth cleaning habits, i.e., how many times a day they cleaned their teeth.

DNA was isolated from the sample by using the standard protocol of QIAmp (Qiagen Ltd, California, USA). Gingival samples were amplified in duplicates using universal primers targeting the V3–V4 regions of the prokaryotic bacterial 16S rRNA gene: the forward primer with adapter was 341F TCG TCG GCA GCG TCA GAT GTG TAT AAG AGA CAG CCT ACG GGA GGC AGC AG (*Muyzer, De Waal & Uiterlinden, 1993*), and the reverse primer with adapter was R806 GTC TCG TGG GCT CGG AGA TGT GTA TAA GAG ACA GGG ACT ACH VGG GTW TCT AAT (*Caporaso et al., 2011*). PCR reaction (25 µl) was carried out with 2.5 µl genomic DNA, 2x KAPA HiFi HotStart ReadyMix (Kapa Biosystems, USA), and the 0.2 mM forward and 0.5 mM reverse primer modified protocol of Takahashi and colleagues (*Takahashi et al., 2014*). Briefly, denaturation at 95 °C for 3 min, followed by 35 cycles of denaturation at 95 °C for 20 s, annealing beginning at 65 °C and ending at 55 °C for 15 s, and extension at 72 °C for 30 s. The annealing temperature was lowered 1 °C every cycle until reaching 55 °C, which was then used for the remaining cycles. Final elongation was for 5 min at 72 °C. Amplicon purification, second PCR, normalization, pooling, and sequencing was performed with Illumina's 16S metagenomic sequencing library preparation protocol (Illumina Inc., San Diego, California USA). MiSeq®Reagent Kit v3 for 600 sequencing cycles (Illumina Inc. San Diego, California) was used for MiSeq library with a final concentration of 4 pM and with 10% PhiX control. The mock community was a DNA pool composed of *Streptococcus mitis* ATCC 49456, *Streptococcus sanguinis* ATCC 10556, *Streptococcus gorgonii* ATCC 10558, *Aggregatibacter actinomycetemcomitans* ATCC 700685, *Porphyromonas gingivalis* ATCC 33277, and *E. coli* ATCC 25922 (LGC Standards, Teddington, Middlesex, UK).

## Data analyses

The number of patients was 60. After amplification in duplicates, the quality of 63 (out of 120 possible) samples was sufficient for sequencing. After sequencing 34 samples from 23 patients were of sufficient quality for the analyses. Eleven patient samples contained duplicates and 12 samples were single samples. The replicates were averaged before statistical analyses. To investigate the gingival pocket bacterial communities of the study patients and to assess the alpha diversity (chao1-index) (*Chao, 1984*), we used the data analysis protocol presented in Fig. 1 in which we show the use of the softwares QIIME (*Caporaso et al., 2010*), trimmomatic (*Bolger, Lohse & Usadel, 2014*), phyloseq

Illumina MiSeq raw files (demultiplexed)

1. Quality Control (QC)

> a. QC check with FastQC
> b. Trim primers
> c. Trim low quality bases (phred score <20)
> d. Remove short sequences (<200 base pairs)
> e. QC check with FastQC

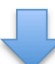

2. Data preparation: Prepare input files for QIIME

> a. Merge paired ends using fasta-join in QIIME
> b. Generate QIIME sequence and map file

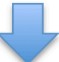

3. Contamination filtration

> a. Remove chimera using usearch61 implemented in QIIME
> b. Remove mitochondrial, chloroplast, archaea and eucaryotic sequences using mothur functionalities
> c. Remove bacterial contamination using control and replicate sequences

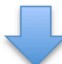

4. OTU clustering

> OTU clustering (97% sequence similarity) with uclust in QIIME

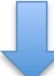

5. Taxonomy assignment

> Assign taxonomy to representative OTU sequences using Human Oral Microbiome Database (HOMD) with uclust method implemented in QIIME

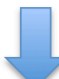

6. Downstream analysis

> a. Alpha diversity analysis with phyloseq
> b. Differential abundance analysis (DAA) using DESeq2 implemented in phyloseq

**Figure 1 Data analysis protocol.** Sequences were clustered into the operational taxonomic units (OTUs) using 97% similarity threshold.

(*McMurdie & Holmes, 2013*) and DESeq2 (*Love, Huber & Anders, 2014*). HOMD (*Chen et al., 2010*) was used as a reference database. The regression analyses were made with and without covariants (sex and age).

Statistical analyses were performed using R software (*R Core Team, 2013*).

## RESULTS

The major phyla detected from the gingival pocket crevicular fluid samples were *Firmicutes* (40.1%), *Bacteroidetes* (28.0%), and *Fusobacteria* (10.3%). The major classes were *Bacteroidia* (23.2%), *Negativicutes* (21.3%), *Clostridia* (11.2%), and *Fusobacteriia* (10.3%).

The association of tooth brushing to the alpha diversity (chao1 index) level of gingival pocket microbiome was calculated. The patients who reported brushing their teeth twice a day had the lowest diversity, and those patients who brushed their teeth more seldom had the highest diversity (Fig. 2) ($p = 0.0002$; R2 = 34%; Regression analysis). In addition, we also repeated the same regression analysis with covariates age and sex [p(adj) = 0.09; R2 = 11%]. The relative numbers of phyla divided into groups by the reported tooth brushing frequency are shown in Fig. 3. To see which bacterial taxa significantly differed between the groups, differential abundant analysis (DAA) was performed (Supplemental Information 1). Two fusobacteria (both *Leptotrichia* -genera) and one bacteroidetes (*Prevotella* -genus) reduced significantly along with teeth brushing ($p = 0.0001$, $p = 0.0007$, and $p = 0.004$, respectively; Fig. 4 and Supplemental Information 1).

## DISCUSSION

The effect of tooth brushing on the alpha diversity level of the gingival pocket microbiome has previously been studied by Do Nascimento's (*Do Nascimento et al., 2015*) and by Chen's (*Chen et al., 2011*) groups. Do Nascimento's group used the DNA checkerboard hybridization method to show that toothbrush bristles impregnated with silver nanoparticles reduced the total and individual genome count in the supra- and subgingival plaque biofilm after 4 weeks of brushing. However, chlorhexidine was not found to be effective in reducing the total genome counts in both supra- or subgingival biofilm after 4 weeks of brushing (*Do Nascimento et al., 2015*). Chen and colleagues suggest that a favorable alteration in the microbial composition of dental plaque occurs 24 h after tooth brushing and professional dental prophylaxis. Importantly, an increase in bacterial diversity was observed after 24 h (*Chen et al., 2011*). In addition, Takeshita also showed that low bacterial diversity is associated with better oral health (*Takeshita et al., 2016*). Our results are in line with these previous findings.

In a healthy gingival pocket, there is a sparse streptococci dominating microbiome, into which different gram-negative rods (prevotella, fusobacteria, and porphyromonas) start colonizing if the plaque is left undisturbed (*Marsh & Martin, 2009*). Takahashi and colleagues have proposed that fusobacteria are one of the key organisms in the early stage and create favorable conditions for other periodontal bacteria and subgingival inflammation through its allogenic factors, such as pH, oxygen levels, temperature, osmotic pressure and oxidation–reduction potential, and protein-based nutrients. Fusobacteria are

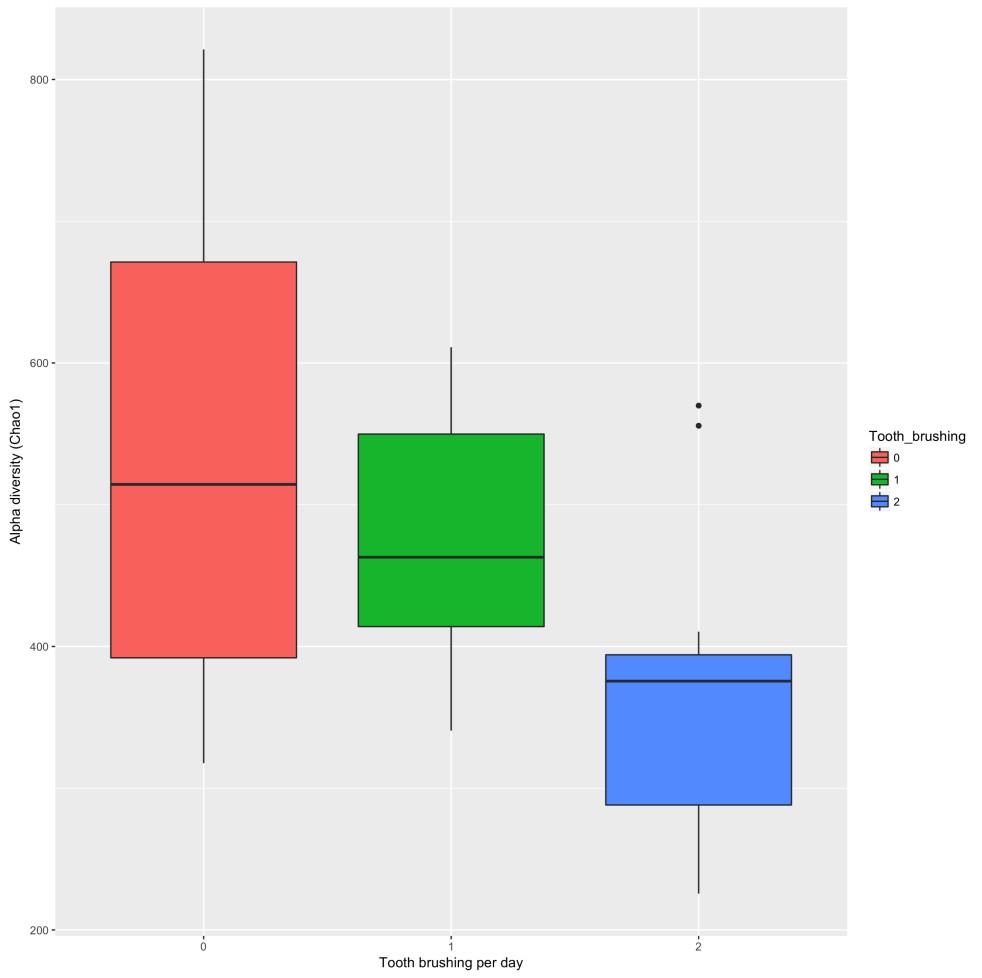

**Figure 2** **The effect of tooth brushing to the alpha diversity (chao1 index) level of gingival pocket microbiome.**

thus known to have a key role in pathogenic subgingival biofilm formation and the binding of other colonizers (*Takahashi, 2005*; *Kolenbrander et al., 2006*). Subgingival biofilm may further induce destructive periodontitis and might also allow bacteria to enter the bloodstream and cause infections elsewhere in the body (*Socransky et al., 1998*; *Zhan et al., 2016*). In this study, the increase in the relative number of fusobacteria was seen to be associated with a low frequency of tooth brushing suggesting that tooth brushing may disrupt the maturation process of the forming plaque biofilm.

There is a link between dental infections and cardiovascular diseases (*Desvarieux et al., 2005*; *Liljestrand et al., 2015*; *Buhlin et al., 2011*; *Tonetti & Van Dyke, 2013*). In our previous studies, we detected both dental bacterial DNA and a bacterial driven inflammation in intracranial aneurysm tissue samples (*Pyysalo et al., 2013*). *Fusobacterium nucleatum* was shown to be among the most common bacteria in the aneurysm samples (*Pyysalo et al., 2016*). In this study, we have shown that reported frequency of tooth brushing was associated with decreased bacterial diversity and an abundance of fusobacteria in the gingival pocket
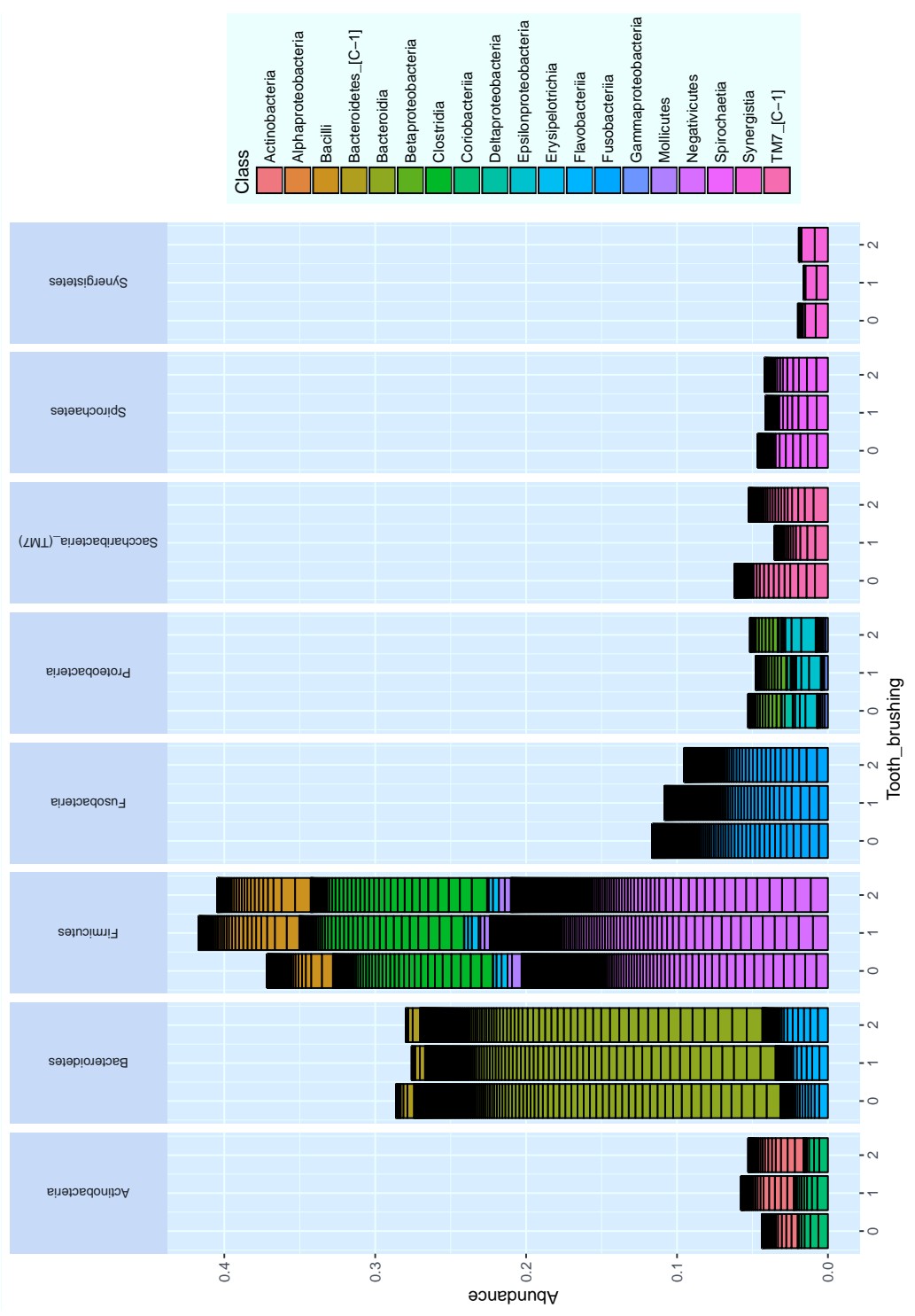

**Figure 3** The relative abundance of taxa at phyla and class levels divided into groups by tooth brushing frequency.

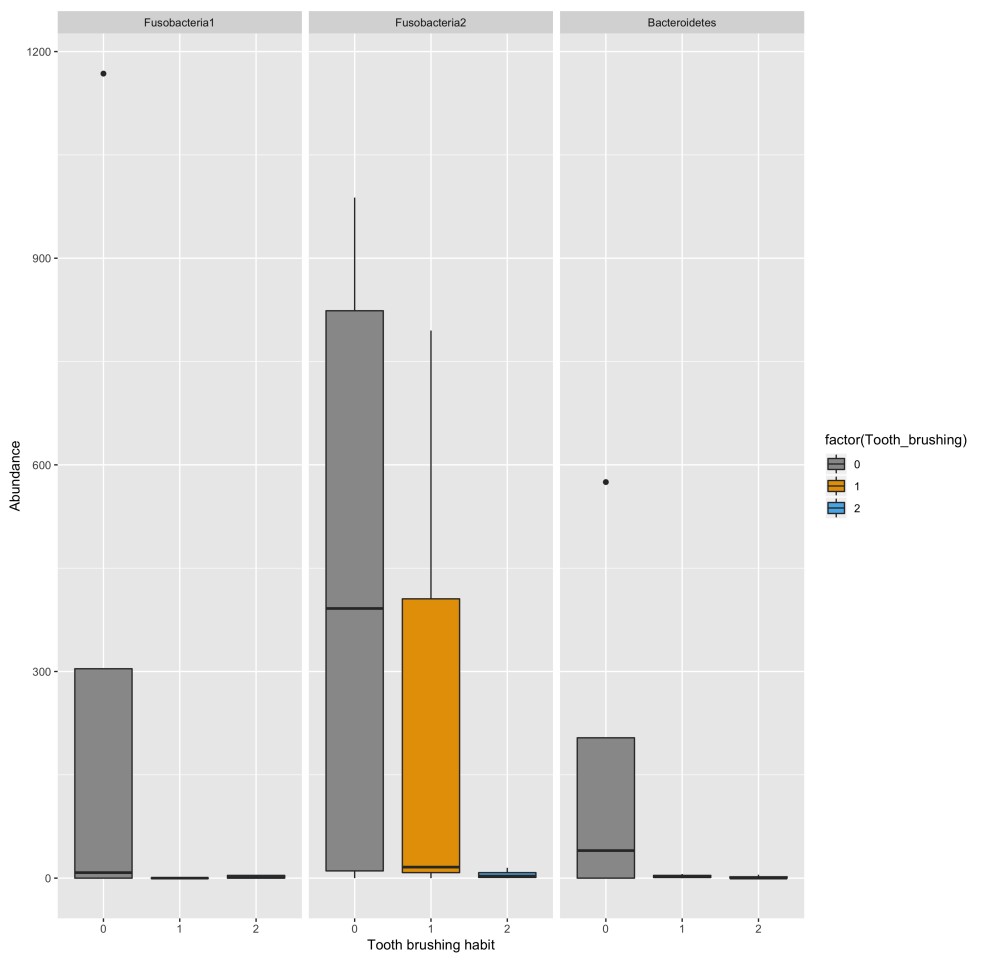

**Figure 4** **Three most significantly differing Operational taxonomic units (OTUs) between the tooth brushing frequency groups.** Fusobacteria1: genus1: *Leptotrichia* Fusobacteria2: genus2: *Leptotrichia* Bacteroidetes: genus: *Prevotella*.

crevicular fluid suggesting that tooth brushing should be strongly recommended, especially for high-risk patients.

The use of molecular methods, such as 16S rRNA gene amplification based, whole genome sequencing, and DNA-DNA hybridization, has revealed fundamental taxonomic changes especially in anaerobic oral bacteria, such as fusobacteria, due to their slow/poor cultivation or none cultivation properties. Although 16S rRNA gene sequencing is a novel technique in the identification of cultivation-independent bacterial communities and remains the gold standard in the determination of bacterial diversity, it has several limitations. For example, it is not the recommended method in species-level identification due to the sequence similarity in the 16S gene. Most bacterial strains have 97% 16S rRNA sequence similarity, suggesting that 16S rRNA gene sequencing provides genus identification in most cases (>90%) but only 65% to 83% species information, leaving between 1% and 14% of the isolates unidentified. Moreover, the obtained sequences

are clustered and similarity threshold to cluster sequences into 'Operational Taxonomic Units' (OTU) is usually 97%, and therefore limits the sequence data (*Stackebrandt & Ebers, 2006*; *Janda & Abbott, 2007*; *Petti, 2007*; *Nguyen et al., 2016*). Due to these limitations, we identified OTUs into phylum and genus, not species levels. In addition, it would be informative to conclude something on cause–effect in high-risk patients. The cross sectional nature of the data in this study does not, however, support a direct cause–effect relationship. To study the cause–effect of tooth brushing on the bacterial profile, we would need a longitudinal study design. We also used sex and age as covariates in the regression model. The addition of covariates to the model however worsens the model fit as shown by the higher *P*-value for adjusted R squared. Due to the questionnaire based nature of the assessment of confounding factors other than age and sex, we did not consider collecting other information reliable enough. Other possible confounding factors could be dietary related factors, the delay between latest tooth brushing and sampling, and the frequency of using mouthwash, for example. The results of this study support the hypothesis that the increase in tooth brushing frequency has a positive effect on subgingival pocket microbiome, which could be considered as a long term effect. Short term effects on supragingival plaque can be caused by dietary factors, such as high sugar intake and occasional use of mouthwash. However if tooth brushing frequency remains constantly high enough, the bacterial plaque has a limited chance to invade the subgingival space.

## CONCLUSION

Subgingival microbiota plays an important role in the pathogenesis of gingival inflammation. The collection of non-invasive gingival crevicular fluid (GCF) and the identification of bacterial communities in it may provide an informative diagnostic tool to evaluate the microbiota environment associated with deep pocket depths and severe gingival inflammation in patients undergoing preoperative dental screening. Subgingival bacterial diversity and the relative abundance of fusobacteria in GCF may be reduced by active tooth brushing.

### Funding

The study has been financially supported by the Jane and Aatos Erkko Foundation (Tanja Pessi), the Competitive State Research Financing of the Expert Responsibility area of Tampere University Hospital (Terho Lehtimäki, Pekka Karhunen and Tanja Pessi), the Academy of Finland (Grant no. 286284 for Terho Lehtimäki), the Tampere Tuberculosis Foundation (Terho Lehtimäki, Pekka Karhunen,Tanja Pessi), the Academy of Finland (grant 286284 for Terho Lehtimäki), the Laboratory Medicine Foundation (Kati Sundström, Tanja Pessi), and the Finnish Clinical Chemistry Foundation (Tanja Pessi). The funders had no role in study design, data collection and analysis, decision to publish, or preparation of the manuscript.

## Grant Disclosures

The following grant information was disclosed by the authors:

Jane and Aatos Erkko Foundation (Tanja Pessi).

Tampere University Hospital (Terho Lehtimäki, Pekka Karhunen and Tanja Pessi).

Academy of Finland: 286284.

Laboratory Medicine Foundation (Kati Sundström, Tanja Pessi).

Finnish Clinical Chemistry Foundation (Tanja Pessi).

## Competing Interests

Terho Lehtimäki, Pekka Karhunen, Kati Sundström and Pashupati Mishra are employees of Fimlab Laboratories Ltd. The authors declare there are no competing interests concerning this study.

## Author Contributions

- Mikko J. Pyysalo conceived and designed the experiments, analyzed the data, prepared figures and/or tables, authored or reviewed drafts of the paper, approved the final draft.
- Pashupati P. Mishra conceived and designed the experiments, performed the experiments, analyzed the data, prepared figures and/or tables, authored or reviewed drafts of the paper, approved the final draft.
- Kati Sundström performed the experiments, authored or reviewed drafts of the paper, approved the final draft.
- Terho Lehtimäki and Pekka J. Karhunen contributed reagents/materials/analysis tools, authored or reviewed drafts of the paper, approved the final draft.
- Tanja Pessi conceived and designed the experiments, performed the experiments, analyzed the data, contributed reagents/materials/analysis tools, prepared figures and/or tables, authored or reviewed drafts of the paper, approved the final draft.

## Human Ethics

The following information was supplied relating to ethical approvals (i.e., approving body and any reference numbers):

The Pirkanmaa hospital district ethics committee granted ethical approval to carry out the study (code R12217).

## Data Availability

Pyysalo Mikko, Mishra Pashupati P., Sundström Kati, Lehtimäki Terho, Karhunen Pekka J., & Pessi Tanja. (2018). Fluid samples from gingival pockets [Data set]. Zenodo. http://doi.org/10.5281/zenodo.1252799.

## Supplemental Information

Supplemental information for this article can be found online at http://dx.doi.org/10.7717/peerj.6316#supplemental-information.

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
