# Peer review of "Increased tooth brushing frequency is associated with reduced gingival pocket bacterial diversity in patients with intracranial aneurysms"

_PeerJ, doi:10.7717/peerj.6316_

## Round 0.1 · original submission · Major Revisions

Please address the reviewers' concerns. In particular, the reviewers bring up some major concerns with regard to experimental design and associated conclusions and these should be addressed. In addition, the availability of the data was brought up as an important issue with regard to reproducibility.

Reviewer 1 ·

Basic reporting

The sequence data was not available.
That is a road block to replicate the study.

Experimental design

The design is questionable, as the cohort is a group of rare tumor patients. How the conclusion from the cohort is generalizable is unaddressed.
The statistical methods used did not adjust for likely confounding factors, such as common demographics, prior periodontal condition, dietary and etc.

Validity of the findings

While tooth brushing is generally believed to drop diversity temporarily, it is believed to be of modest effect (Hall et al.
Inter-personal diversity and temporal dynamics of dental, tongue, and salivary microbiota in the healthy oral cavity). Since the current single time point design did not address the variability of temporal diversity change, it is hardly conclusive.

Moreover, the use of a group of rare tumor patients instead of dental patients and healthy population makes the conclusion less generalizable.

Additional comments

Given the design flaw, the article would be hardly revisable for the purpose of current conclusion. My suggestion is, the authors may find interesting points from reanalyzing their microbiome data together with the patients' tumor conditions and wrapping their data around these points.

Reviewer 2 ·

Basic reporting

The manuscript is written in a clear and unambiguous English.

Please make the raw data available. If possible, it would be excellent to have a self-containing script that fully reproduces the results and figures as well, together with the raw data.

Also, please properly cite all the software packages you used (e.g. QIIME, DESeq2, Trimming software, phyloseq, ...?!). Also, please refer to the used methods and statistics that are calculated (e.g. Chao1).

When it comes to the Figures, Figure 2 needs some improvements, if the abundances are merged on class level, it is not clear to me why there are more bars of the same colour? And how to interpret the parts that are black/borders? It might be worth trying out a barplot?!

In Figure 3, to me, it seems that there are more than 22 samples? Could you clarify, why there are so many dots? I suppose they should indicate the samples? If they were used also in the regression analysis, is the analysis based on the 22 samples or the ones indicated in Figure 3?

Also in Figure 4, I am not sure about the boxes of same colour, how to interpret theses. Could you please clarify this?

Experimental design

The research question is well defined. However, I have some doubts about the experimental design itself. The authors find significant differences in the gingival pocket bacterial diversity associated with the increased tooth brushing frequency but ignore the large amount of other, possible confounding factors that might need to be discussed and/or taken still into consideration, if applicable. Possible factors are

1. Is there any effect on the bacterial diversity for the time between sampling and the last time brushing the teeth? This time would be also expected to be smaller for people who brush twice a day and hence, is potentially very influential.

2. What is the effect of using mouthwash? My assumption is, people who brush more often the teeth have also the tendency to use more often mouthwash.

3. How long time before the sampling, patients ate something?

Please discuss the confounding factors as possible effects.

Also, you mentioned that the probes were taken from the deepest gingival pocket. How was this depth distributed among the three brushing groups? There certainly is an influence of periodontal probing depth on the gingival bacterial communities and hence should be similar among the groups to compare.

Further, you mention that 38 samples have been discarded from the analysis due to insufficient quality. Could you please provide a clear justification, what criteria they failed to meet and how the 22 remaining samples passed the check? Maybe add the quality checks to the supplement (maybe combined using e.g. MultiQC, if they failed on the raw data level)

Validity of the findings

The Chao1 index has (like any other Alpha diversity index) its limitations and there are sometimes large differences between Chao1 and other, popular indices. In order to declare a strong hypothesis as the authors do, I would like to see also other, commonly used indices applied to the data and see, if they support the hypothesis, also.

---

## Round 0.2 · Major Revisions

The reviewers still have significant concerns, especially with regard to the study design. Questions regarding independence of the samples and repeated measurements should be thoroughly addressed or the approach appropriately modified.

Reviewer 1 ·

Basic reporting

From the rebuttal letter, my concerns were acknowledged but mostly unaddressed. The authors did release their data, but they used a very uncommon data website and I am surprised neither NCBI or ENA was used.

Experimental design

The questioning of their design remains. The authors claim they do not want to generalize the finding but they forgot to quantify their title with the cancer types, which became very misleading if people just take the face value.

Validity of the findings

With the limited design, also remained are likely confounding factors, as also pointed out by Review #2. If the authors cannot release meta data, to reach a conclusion trustable, at least summary statistics and p-values from multivariate models should be reported, rather than pure microbiome data.

Reviewer 2 ·

Basic reporting

Not considered, see below.

Experimental design

After seeing the clinical information of the included samples, another issue in the experimental design became obvious. There are repeated measurements in the data, some of the included individuals have been sampled repeatedly and consequently have been included also repeatedly into the analysis.

Validity of the findings

The used methods rely on an iid assumption of the samples, which is, however, violated by including repeated measurements without taking this into account for the analysis. Please, either take the information of repeated measurements into account (=adjust the used methods accordingly), or remove the repeated measurements from the analysis.

In my opinion, using a few repeated measures biases the results tremendously and consequently the results are in their current form not valid.

Additional comments

You mention that you describe the samples now more detailed, but it was still not obvious that you were using repeated measurements in the data. As written above, either justify repeated measurements or remove them

---

## Round 0.3 · Major Revisions

Thank you for addressing reviewer concerns. I think this manuscript is actually quite close and not necessarily having major issues if just a couple remaining reviewer issues can be addressed. In particular, there still appears to be issues with regard to the analysis between frequency of toothbrushing and microbial diversity, with discrepancies between discussed and shown sample sizes. For instance, it is not clear how many samples are used to support the conclusions that are made. This should be clarified as should the figure issues brought up by the reviewer. In addition, a discussion of confounding factors would be extremely helpful. Thank you for your previous efforts in improving the manuscript.

Reviewer 2 ·

Basic reporting

I am still not sure about your reported sample sizes, sorry! In the abstract you write that you started with n=60 patients, but then later in the the M&M part you tell that you use 34 samples from 23 patients (out of 63 samples in total). Those numbers still do not seem to match, if you have n=60 patients in the beginning and from there then 63 samples, how can you have for 11 patients even two samples? Please, still go once more carefully over your sample size description. Also, the amount of observations (=dots) in Figure 2 does still not match the description in the M&M part. If you took in total 34 samples from 23 patients and you averaged the repeated measurements, there should be only 23 dots in that figure, if I understood it right?!! And as your regression analysis is based on this data, are you sure you applied the regression analysis on the averaged data?

Experimental design

As pointed out already in the previous review by myself and also by reviewer #1, there are conceptual problems in the experimental design as many potential confounding factors have not been controlled. Of course that is nothing that could be changed at this stage anymore, but I still miss a critical discussion of this in the manuscript. You write that you findings "are in line with these previous findings (line 159)" (e.g. that bacterial diversity increases 24h after tooth brushing) but at the same time you have not asked the patients when they brushed their teeth last time. I believe your information of 0,1,2 brushings per day is the normal behaviour, but people might deviate from that on some days (or some brush only in the evening others in the morning).

Validity of the findings

I do not doubt that there is a correlation between the frequency of tooth brushing and the microbial diversity. The problem that I still have with this study is that you do not discuss the possible confounding factors (even after having them pointed out). Especially, in the context of missing confounding factors, please be cautious about mixing correlation and causation. (Silly example, there is a strong correlation between sleeping with leather shoes still on the feet and waking up with a headache. The conclusion would be, sleeping with leather shoes causes headache. The obvious underlying causative factor is, however, rather another one...).

That said, I think, you really should discuss still other possible effects that could cause your observation and not only bring it down to the frequency of tooth brushing.

---

## Round 0.4 · accepted · Accept

Thank you for addressing the reviewer concerns and updating the relevant manuscript sections. Congratulations again.

#